# Point Cloud Registration Based on Fast Point Feature Histogram Descriptors for 3D Reconstruction of Trees

Yeping Peng [1,2], Shengdong Lin [1,2], Hongkun Wu [3] and Guangzhong Cao [1,2,*]

1    College of Mechatronics and Control Engineering, Shenzhen University, Shenzhen 518060, China; yeping.peng@szu.edu.cn (Y.P.)
2    Guangdong Key Laboratory of Electromagnetic Control and Intelligent Robots, Shenzhen University, Shenzhen 518060, China
3    School of Mechanical and Manufacturing Engineering, The University of New South Wales, Sydney, NSW 2052, Australia
*    Correspondence: gzcao@szu.edu.cn

**Abstract:** Three-dimensional (3D) reconstruction is an essential technique to visualize and monitor the growth of agricultural and forestry plants. However, inspecting tall plants (trees) remains a challenging task for single-camera systems. A combination of low-altitude remote sensing (an unmanned aerial vehicle) and a terrestrial capture platform (a mobile robot) is suggested to obtain the overall structural features of trees including the trunk and crown. To address the registration problem of the point clouds from different sensors, a registration method based on a fast point feature histogram (FPFH) is proposed to align the tree point clouds captured by terrestrial and airborne sensors. Normal vectors are extracted to define a Darboux coordinate frame whereby FPFH is calculated. The initial correspondences of point cloud pairs are calculated according to the Bhattacharyya distance. Reliable matching point pairs are then selected via random sample consensus. Finally, the 3D transformation is solved by singular value decomposition. For verification, experiments are conducted with real-world data. In the registration experiment on noisy and partial data, the root-mean-square error of the proposed method is 0.35% and 1.18% of SAC-IA and SAC-IA + ICP, respectively. The proposed method is useful for the extraction, monitoring, and analysis of plant phenotypes.

**Keywords:** point cloud registration; fast point feature histogram; Bhattacharyya distance; 3D reconstruction

## 1. Introduction

Plant phenotype features, such as plant height, canopy size, and diameter at breast height, can provide substantial information on plant growth status, whereby breeding programs and conservation strategies are aimed at improving crop yield and quality. Given its high resolution, light detection and ranging technology (LiDAR) is being widely used to obtain various plant phenotypes, such as the leaf structure of maize [1], the height of peanut seedlings [2], and the biomass of sugarcane [3]. However, color information is unavailable. To monitor plant phenotype visually, computer vision has emerged as an alternative sensing strategy to monitor plant growth status [4].

Vision-based 3D reconstruction technology has been widely used in the fields of mechanical, agricultural, industrial, and urban planning applications [5–7]. Specifically, the 3D modelling of a plant's image reconstruction algorithm holds great significance for the visual monitoring of plants and their phenotype measurements. However, vision-based plant phenotype monitoring is still noted with specific limitations for different camera systems. Three-dimensional (3D) cameras, such as the time-of-flight camera and the binocular camera, are used to collect plant phenotype features [8,9], and have been proven effective in indoor scenes. The single RGB (color components of red, green, and blue) camera is one of the most popular sensors due to its low cost and low weight [10] and has

been utilized to estimate the height and canopy coverage of wheat, rice, and maize [11–13]. A point cloud describes the geometric structure and morphological traits of the object. This information is suitable to extract plant phenotype parameters. Compared to traditional manual measurement, 3D point cloud visualization offers advantages such as it not being labor intensive, and it is non-destructive and objective. Compared with RGB images, point cloud also has more comprehensive 3D information [14,15].

Studies on 3D reconstruction have been conducted for short plants. For example, a multi-view stereo (MVS)-based 3D reconstruction was developed to acquire 3D point clouds of corns by collecting image sequences of branches and leaves [16]. This method can reconstruct the morphological structure of plants, but it requires placing the plants on a specific platform. Li et al. [17] took images of six short plants within the local range of a binocular camera and reconstructed the plant models with a cost measure that combines the absolute differences and the census transform. Additionally, 3D models of wheat and rice were established using RGB images [18]. We observed that most of the existing measurement/sensing techniques are suitable for small plants, and there are relatively few studies that use a single device for complete reconstructed point clouds of large plants. However, for tall plants, such as agricultural and forestry trees, in practical scenarios, it is still a challenging task from a single device [19]. A terrestrial device can capture more trunk information and texture, but the canopy information is not enough. To address this issue, an unmanned aerial vehicle (UAV) can be applied with its flexibility in the air and can acquire the information of the canopy top. However, due to limitations in camera rotation angle and collision safety range, the trunk images captured by UAVs are often occluded and incomplete. As a result, to obtain a set of more complete point clouds, the point clouds captured by these two methods shall be aligned—namely, point cloud registration. The goal of point cloud registration is to find a transformation that aligns the source point cloud with the target point cloud.

Most of the traditional methods achieve point cloud registration by coarse matching first and then fine matching. One of the most common strategies is the combination of sample consensus initial alignment (SAC-IA) and iterative closest point (ICP) [20]. SAC-IA firstly samples multiple points from the source point cloud, then identifies a subset of similar points in the target point cloud, and finally selects a point from the subset of similar points as corresponding points. This selection is repeated to optimize the transformation with the least error. However, it is difficult to obtain the global optimum for point cloud registration with a low overlap rate due to the larger number of outlier points. This limitation can be overcome by feature extraction and matching. ICP iteratively associates the point pair with minimum Euclidean distance, and the overall distance is minimized with the least squares method, where the transformation matrix is estimated. However, a proper initialization is needed, and local optimum is another potential concern. How to ignore the influence of data initialization on point pair matching becomes a research topic. In addition, many registration methods based on deep learning have been proposed recently. They have advantages in some respects, such as the accuracy achieved by specific experimental scenarios. PointNet [21] and PointNet++ [22] are widely used as end-to-end point cloud learning frameworks. 3Dmatch [23] establishes correlation between 3D data by learning descriptors. An unsupervised point cloud registration method [24] can leverage differentiable alignment and rendering to learn point cloud registration from RGB-D videos. However, compared with traditional methods, deep learning requires a more expensive computing platform and a large amount of training data. In practice, there is still a considerable gap in this approach to meet the needs of real-time online monitoring.

The registration performance of tree point clouds captured by airborne and terrestrial platforms depends on the amount of partial overlap. However, the areas where these two point clouds overlap tend to be small. For the data with insufficient overlap, it is difficult for the typical SAC-IA to obtain a good solution consistent with the data due to the relatively few inliers in the matching pair [25]. This problem will affect the point cloud registration results, particularly in complex and large-scale scenes. Furthermore, if coarse registration

fails to provide a good initial pose, the subsequent fine matching also tends to give a limited performance. To mitigate the limitations imposed by initial pose requirements, especially for the registration under the low overlap, this paper presents an improved registration method based on a fast point feature histogram (FPFH). Specifically, an FPFH is constructed to describe the local point cloud features according to the estimated normal vector and spatial relationship within the neighborhood point's subset. Its translation and rotation invariant properties make the description more consistent. As a common similarity metric, the Bhattacharyya distance is applied to define a correlation metric and select matching pairs of points between two point clouds. In addition, a hybrid selection strategy that combines the scoring rules and random sample consensus algorithm (RANSAC) is proposed to select more reliable matches. Finally, a transformation matrix is solved by singular value decomposition (SVD). The highlights of this work are summarized as follows:

(1) An association description and assessment method, based on an FPFH and the Bhattacharyya distance, is proposed. This method is rotationally invariant to point clouds and is more accurate for point clouds with noise and low overlap.

(2) A matching pair pruning strategy with RANSAC is developed. By evaluating the correlation between matching pairs, the erroneous matches are removed more effectively, leading to an increased proportion of correct matching pairs.

(3) A series of registration experiments on real-world tree point clouds and synthesized point clouds with Gaussian noise demonstrate the effectiveness of the proposed method.

The rest of this paper is organized as follows. The framework of the plant phenotype monitoring system is introduced in Section 2. The method proposed and its mathematical model are given in Section 3. In Section 4, experimental results on real-world point clouds of agricultural and forestry trees are presented and discussed. Finally, the conclusion of this paper is given in Section 5.

## 2. Phenotypic Structure Monitoring System for Tall Plants

The hybrid phenotypic structure monitoring system of trees encompassing a UAV and a terrestrial acquisition platform is shown in Figure 1. For agricultural and forestry trees, canopy data are acquired by a low-altitude remote sensing platform (UAV, DJI MAVIC 2) [26]. However, due to the limited flight range and blind zone of the camera, it is not a trivial task to capture complete images of the tree trunk. The terrestrial acquisition platform (RGB camera) is capable of capturing more detail of the tree trunk texture. Therefore, the image capture capabilities of these two platforms are combined to obtain a comprehensive image of the trees. Finally, the 3D reconstruction based on the image sequence is conducted to construct the point cloud model of the canopy and trunk. Those two point clouds are converted into the world coordinate for registration and result in the generation of a 3D model encompassing the entire tree.

As can be observed in Figure 1, the reconstructed point cloud of the canopy lacks tree trunk information. This is due to partial occlusion of the tree trunk by the canopy, making it challenging to reconstruct. However, this issue can be addressed by incorporating the terrestrial imaging platform because it mainly captures the image of the branches. By examining the point cloud of the tree trunk, more details in the trunk part are obtained. It shall also be noted that there is quite a considerable overlap between the aerial and terrestrial point clouds, which is regarded as the key feature to combining those two datasets. Specifically, by extracting similar geometric features, point cloud alignment is feasible, which allows the reconstruction of the 3D model of the whole tree. This provides an important path for the extraction of the plant phenotype.

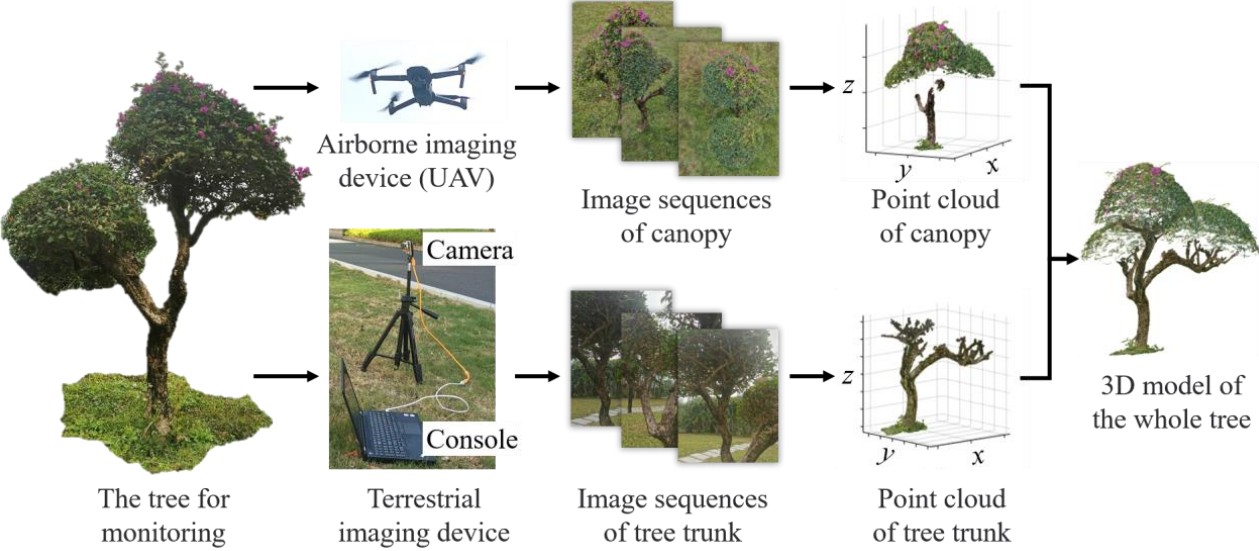

**Figure 1.** The phenotypic structure monitoring system of trees combined with the UAV and terrestrial capture platform.

## 3. Improved Point Cloud Registration Method Based on FPFH

After reconstructing the point clouds of the canopy and trunk using different visual platforms, feature extraction and registration are performed to reconstruct the complete plant model. This process is shown in Figure 2. The key steps are: (1) The point cloud data are preprocessed to eliminate the outliers caused by background and noise. This step not only suppresses the disturbance but also reduces the data size, which improves both accuracy and efficiency of subsequent point cloud processing. (2) The 3D feature descriptor (FPFH) of the point cloud is calculated to characterize the local geometric features of the spatial points. (3) The correlation between the canopy and trunk points is evaluated via the Bhattacharyya distance, and the matching point pairs are identified according to the configured correlation rules. (4) The point cloud data are sampled based on the correlation level and the distance distribution. Subsequently, a set of matching point pairs is generated. The mismatching point pairs are removed based on RANSAC. (5) The estimated transformation (rotation matrix $R$ and translation vector $t$) between point clouds is solved based on multi-view geometry. Point cloud stitching is then implemented based on the transformation.

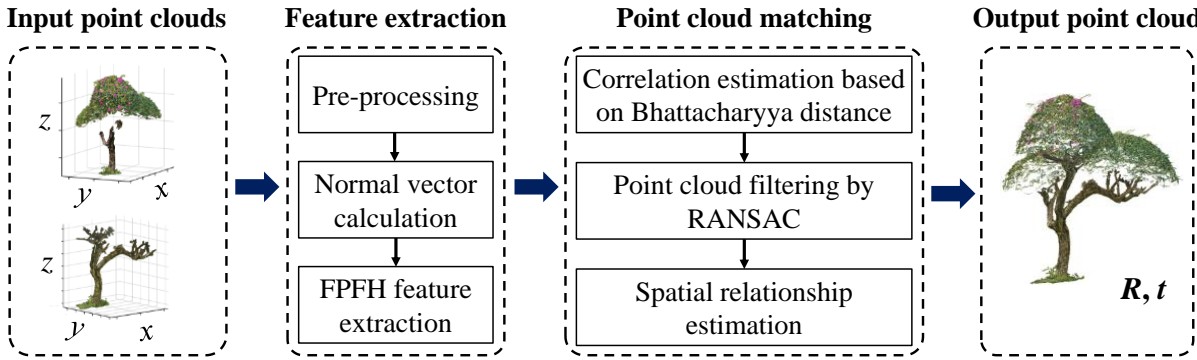

**Figure 2.** The workflow of the proposed point cloud registration of trees.

### 3.1. Point Cloud Data Preprocessing

Point cloud preprocessing includes background removal, denoising, and downsampling. Firstly, the target tree is manually cropped from the original point cloud. Owing to the noises produced by sensors and 3D reconstruction errors, outliers are generally generated and sparsely distributed in the reconstructed point clouds, which may cause mismatching of partial point clouds and result in the failure of registration. To address this problem, a statistical filtering method is used to delete the outliers. First, the average distance of each point in the point cloud to its k-neighborhood points is calculated. Then, the mean $\mu$ and standard deviation $\sigma$ of all average distances are calculated. Based on the Gaussian distribution, most data points are expected to be distributed within a certain range around the mean value, and those data points that significantly deviate from the mean value can be considered as outliers. To identify the outliers, a commonly used method is to assess the average distance of each point to its k-neighborhood neighbors using a global threshold. The threshold $d_{max}$ in statistical filtering method is defined as:

$$d_{max} = \mu + g\sigma, \tag{1}$$

where $g$ is a proportionality coefficient. If a point's average distance of k-neighborhood neighbors is larger than $d_{max}$, the point is labeled as an outlier and discarded. The $g$ value can be determined by the distribution of noise points and the complexity of point cloud, and its empirical value is in the range of (1.0, 3.0). The higher the value, the fewer points will be removed, and more noise will be retained.

In addition, referencing the reported work [27], voxel filtering is adopted to downsample point clouds and decrease the computations. A 3D voxel grid with a given size is first applied to the point cloud. The centroid of all points in each grid is then calculated, and they are used to represent all points in the grid. Thus, the data size of the point cloud is reduced while preserving the initial geometric structure characteristics.

### 3.2. 3D Point Cloud Feature Descriptor Extraction

The good description of 3D point cloud features is conducive to registration. In recent years, a variety of methods have been proposed to improve the accuracy and efficiency of point cloud registration [28]. Specifically, 3D shape context (3DSC) [29] counts and normalizes the points in angular and radial bins of spherical regions around each point, which forms a feature vector that captures the shape and structure of point clouds. However, 3DSC is sensitive to noise [30]. The signature of histograms of orientations (SHOT) [31] descriptor is a variant of 3D shape context which uses histograms to encode the orientation of the normals of the points in each spherical bin. It also integrates multiple cues such as color and texture within a single feature vector, called a signature. SHOT is computationally expensive and requires a large number of bins [32]. A point feature histogram (PFH) [33] uses normal vectors to calculate the relationship between query points and their neighborhood points where the properties of the neighborhood points are represented by histograms. An FPFH [20] is an improved feature description of a PFH. Compared with a PFH, the computational complexity of an FPFH is reduced. In addition, an FPFH can achieve a better balance between descriptiveness and computational efficiency and better distinguish geometric features [34,35]. As depicted in Figure 3, an FPFH shows significant differences in geometric objects such as plane, edge, and vertex, which is conducive for feature matching.

Before extracting FPFH features, it is necessary to compute the normal vector of each point. As shown in Figure 4, $p_i$ $(i = 1, 2, \ldots, n)$ are the neighborhood points of $x$, $n$ is the number of neighborhood points, $p_c$ is the centroid of $p_i$, and $n_p$ is the normal vector of the point $x$ on the fitting plane $L$.

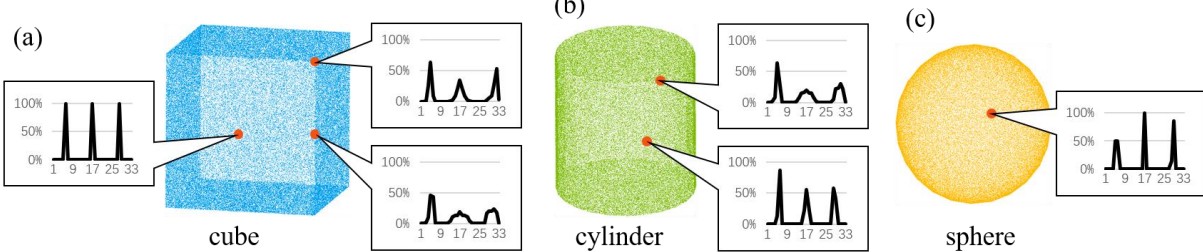

**Figure 3.** FPFH shows significant differences in different geometric features: (**a**) a point cloud of a cube, and the FPFHs of its plane, edge, and vertex; (**b**) a point cloud of a cylinder, and the FPFHs of its edge and surface; (**c**) a point cloud of a sphere, and the FPFH of its surface. The abscissa of the histogram represents the number of points, and the ordinate represents the percentage of points falling in each bin.

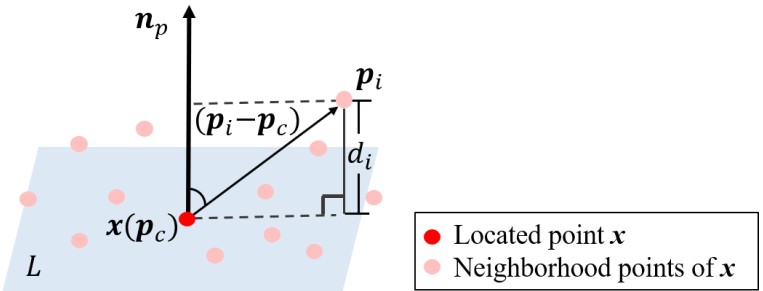

**Figure 4.** Schematic diagram the normal vector of a point $x$.

Firstly, the centroid $p_c$ of a point in the neighborhood of $x$ is calculated as follows:

$$p_c = \frac{1}{n}\sum_{i=1}^{n} p_i,\tag{2}$$

where the distance $d_i$ from a neighborhood point $p_i$ to the plane $L$ is equal to the projection of $(p_i - p_c)$ on the normal vector $n_p$, denoted as:

$$d_i = (p_i - p_c)\cdot n_p.\tag{3}$$

The covariance matrix is constructed according to the neighborhood point $p_i$ and the centroid $p_c$, which is expressed as:

$$C = \sum_{i=1}^{n} (p_i - p_c)(p_i - p_c)^T.\tag{4}$$

The eigenvalues of the covariance matrix are denoted as $\lambda_e$ (assuming $0 \leq \lambda_0 \leq \lambda_1 \leq \lambda_2$) and their corresponding eigenvector $v_e$, $e \in \{0, 1, 2\}$.

According to the theory of principal component analysis and the geometric interpretation of eigenvectors, $v_2$ and $v_1$ carry the majority of the information of the neighborhood data. For planar local point clouds, $v_2$ and $v_1$ span the local plane [36]. Since the eigenvector is an orthogonal basis set, the eigenvector $v_0$ with the smallest eigenvalue is approximately parallel to the surface normal $n_p$ or $-n_p$. By setting a viewpoint $V_p$ as the judgment basis to solve the ambiguity in the normal vector, the specific discriminant is:

$$n_p\cdot(V_p - p_i) > 0.\tag{5}$$

The FPFH can be obtained after the normal of each point is computed. As shown in Figure 5, the initial neighborhood (first-order neighborhood) of point $p_q$ is set as a sphere of radius $r_f$. In order to efficiently extract the neighborhood information, the space with radius $r_f$ of all points in the first-order neighborhood is set as the second-order

neighborhood. Then, the angular and positional relationships between all points in the first-order neighborhood and their neighbors are used to calculate FPFH.

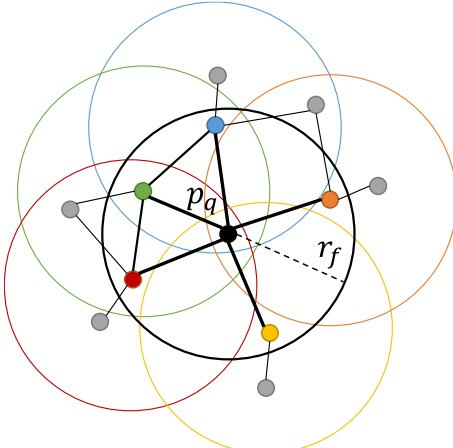

**Figure 5.** The spatial neighborhood of point $p_q$ for the calculation of its FPFH features. The dots in different colors are the centers of the corresponding spherical spaces.

In order to characterize the spatial relationship of the point pairs, a Darboux coordinate system $u$-$v$-$w$ corresponding to the point pairs is established. As shown in Figure 6, $p_s$ and $p_t$ are the source point and target point, respectively, and $\boldsymbol{n}_s$ and $\boldsymbol{n}_t$ are the estimated normal vectors of $p_s$ and $p_t$, respectively. The $u$ axis is consistent with the direction of the normal $\boldsymbol{n}_s$, the direction of the $v$ axis is the direction of the vector obtained by the cross product $(p_t - p_s) \times u$, and the $w$ axis is perpendicular to the $u$-$v$ plane and conforms to the Cartesian coordinate system criterion. It is worth noting that the features based on the local reference frame have translation and rotation invariance.

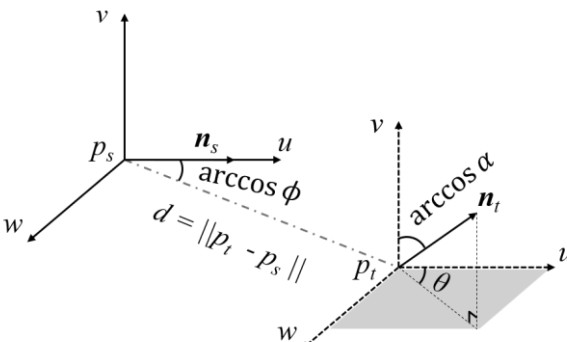

**Figure 6.** The Darboux coordinate system of two points.

The Darboux coordinate system shown in Figure 6 is defined as:

$$\begin{cases} u = \boldsymbol{n}_s \\ v = \frac{p_t - p_s}{\|p_t - p_s\|} \times u \\ w = u \times v. \end{cases} \tag{6}$$

Then, four values are used to describe the relationship between the two points $p_t$ and $p_s$, denoted as:

$$\begin{cases} \alpha = v \cdot \boldsymbol{n}_t \\ \phi = u \cdot \frac{p_t - p_s}{d} \\ \theta = arctan(w \cdot \boldsymbol{n}_t, u \cdot \boldsymbol{n}_t) \\ d = \| p_t - p_s \|, \end{cases} \tag{7}$$

where $\alpha$ is the dot product of the normal vector of $\boldsymbol{n}_t$ and the $v$ axis, $\phi$ is the dot product of the normal vector of $p_s$ and $p_t - p_s$, $\theta$ is the angle between the projection of $\boldsymbol{n}_t$ on the $u$-$v$ plane and the $u$ axis, and $d$ is the Euclidean distance from $p_s$ to $p_t$.

The point cloud density is influenced by the sensor range. Therefore, to mitigate the interference of this factor, the calculation of $d$ is not considered. $\alpha$ reflects the normal relationship between the two points, and $\phi$ is the relative position between two points. Equation (7) shows that $\theta$ is similar to the azimuth of the spherical coordinate system. Furthermore, $\theta$ can be used to describe the orientation relationship between two points. The three variables $(\alpha, \phi, \theta)$ in the FPFH are distributed over as much as possible of the available histogram range without exhibiting bias for specific regions [37].

The calculation of the FPFH for the query point $p_q$ is summarized in the following steps. (1) Calculate $\alpha, \phi, \theta$ of the query point $p_q$ and its k-neighborhood points and denote them as SPFH($p_q$); (2) calculate $\alpha, \phi, \theta$ between $p_q$'s neighborhood points $p_{qi}$ and their respective neighborhood points, denoted as SPFH($p_{qi}$); (3) the three variables $\alpha, \phi, \theta$ are distributed in the histogram. Finally, the SPFH of the query point and the SPFH of the weighted neighbor points are applied in the histogram to obtain the final FPFH via:

$$FPFH(p_q) = SPFH(p_q) + \frac{1}{k}\sum_{i=1}^{k}\frac{1}{\omega_i}SPFH(p_{qi}), \tag{8}$$

where $k$ is the number of points in the neighborhood of $p_q$ and $\omega_i$ is the weight value, which is inversely proportional to the Euclidean distance between $p_q$ and $p_{qi}$. This means that points $p_{qi}$ farther from the query point $p_q$ have less influence on FPFH($p_q$).

The dimension of each feature in the FPFH is set to an odd number to prevent ambiguity caused by noise interference when the angle is $90°$. The three eigenvalues $\alpha, \phi, \theta$ are calculated based on the estimated normal vector. In the noise-free case, the normal vector directions of the non-edge points on the plane should be consistent and perpendicular to the plane. However, in most cases, point cloud data contain noisy points. Hence, the normal vector on the plane point cloud is not strictly perpendicular. Then, $\alpha$ and $\phi$ will have a slight deviation from 90 degrees. If the dimension is even, then two similar points may be divided into two adjacent dimensions separated by 90 degrees due to noise interference, even if their features are sufficiently similar. By setting the dimension to an odd number, this ambiguity can be avoided.

*3.3. Correspondence Estimation of Point Clouds*

After the features are extracted, two point clouds can be associated. In statistics, the distance between different probability distribution functions or histograms is also known as a correlation measure. For two discrete probability distributions $g$ and $h$ in the same domain $X$, the distance measure is widely used as the evaluation index of similarity. Among them, the Bhattacharyya distance, chi-square test, and Kullback–Leibler (KL) divergence are three commonly used distance measurement parameters. However, the chi-square test and KL divergence are both asymmetric, while the Bhattacharyya distance is not only symmetric but also nonnegative and satisfies the triangle inequality. In addition, the Bhattacharyya distance is defined for all possible values of $g$ and $h$, without making any assumptions about the distribution itself. According to the composition characteristics of FPFH features, the correlation metric of an FPFH based on the Bhattacharyya distance is applied to evaluate the difference between the two histograms. The equation of the Bhattacharyya distance is:

$$D_B(g,h) = -ln\left(\sum_{x \in X}\sqrt{g(x)h(x)}\right), \tag{9}$$

where $g(x)$ and $h(x)$ are the values of $g$ and $h$ in the interval $x$. The FPFH is essentially composed of three 11-dimensional histograms that reflect the features of $\alpha$, $\phi$, and $\theta$, respectively. To calculate the correlation of the FPFH, the Bhattacharyya distances of the three histograms are calculated and summated and used as the score of the correlation of

the two FPFHs. According to Equation (9), the correlation between points $p_1$ and $p_2$ can be expressed as follows:

$$C_{FPFH}(p_1, p_2) = D_{B-\alpha}(p_1, p_2) + D_{B-\phi}(p_1, p_2) + D_{B-\theta}(p_1, p_2), \qquad (10)$$

where $D_{B-\alpha}(p_1, p_2)$, $D_{B-\phi}(p_1, p_2)$, and $D_{B-\theta}(p_1, p_2)$ represent the Bhattacharyya distances of $\alpha$, $\phi$, and $\theta$, respectively. $C_{FPFH}(p_1, p_2)$ is non-negative according to the definition of the Bhattacharyya distance. A smaller $C_{FPFH}(p_1, p_2)$ suggests a higher similarity between the two histograms, which also means a higher correlation between the two points. Within the target point cloud, each point in the source point cloud is searching for the point with the least Bhattacharyya distance, and those two points build a matching point pair. The initial set of matching point pairs is composed of these matching point pairs. However, due to the influence of noise and non-overlapping regions, the initial set of matching pairs contains a large number of false matches.

In order to obtain a more correct set of matching pairs, an effective matching pair pruning strategy is needed to remove the false matching. The workflow is shown in Algorithm 1.

---

**Algorithm 1:** Iterative selection of correct matching pairs

---

**Input:** $S_0$: the initial matching pair set; $N$: the number of matching pairs to be selected; $t_e$: the Euclidean distance threshold;
**Output:** $S_{acc}$: the accurate matching pair set;
1:　　Initialization: $S_{acc} \leftarrow \phi$; $n = 1$;
2:　**for** $i = 1 : \text{size}(S_0)$ **do**
3:　　　**if** the point $p_{ti}$ in the target point cloud is selected multiple times,
4:　　　　**then** point $p_{ti}$ reversely selects its most similar point as the
5:　　　　matching pair; delete other duplicate matching pair(s) of point
6:　　　　$p_{ti}$;
7:　　　**end if**
8:　**end for**
9:　　Sort the matching pairs in $S_0$;
10:　Put the first matching pair $p_1$ into $S_1$;
11:　Put the source point of the first matching pair to $p_{tc}$;
12:　**for** $j = 1 : \text{size}(S_0)$ **do**
13:　　　**if** $D_{Euclidean}(p_{tc}, p_{tj}) > t_e$ **then**
14:　　　　$S_1 = S_1 \cup p_j$;
15:　　　　$p_{tc} = p_{tj}$;
16:　　　**end if**
17:　　　**if** $n = N$ **then**
18:　　　　break;
19:　　　**end if**
20:　**end for**
21:　Apply RANSAC to $S_1$, get the inlier set $S_{acc}$.

---

Due to the large data size and small point distance, it is possible that multiple points will be matched to the same point, which is obviously unreasonable. To solve this problem, points that are selected multiple times are allowed to match their most similar points and build new matching pairs. Then, the correlation is scored according to $C_{FPFH}(p_1, p_2)$, and the matching pairs are sorted. By setting a distance threshold, the matching pairs with the highest ranking are selected. The point pairs are distributed wide enough to avoid finding the optimal transformation of the local point cloud. In addition, random sample consensus (RANSAC) is used to reduce mismatched pairs in the set, which removes outliers via repeated model prediction and verification from experimental data. The set of point pairs obtained is then the set of matching pairs with high reliability.

### 3.4. Estimation of Spatial Transformations

The rigid transformation of the two point clouds can be represented by a rotation matrix $R$ and a translation vector $t$. Accurate $R$ and $t$ can make the registration perform well. Singular value decomposition is used to solve the optimal $R$ and $t$ based on the set of matched pairs obtained in the previous section. First, the set of matching pairs is represented as two ordered sets of points:

$$P = \{p_1, \ldots, p_m\}, P' = \{p'_1, \ldots, p'_m\}. \tag{11}$$

The error $e_k$ of the k-th point is defined as:

$$e_k = p_k - (Rp'_k + t), \tag{12}$$

where $p_k$ and $p'_k$ represent the point coordinates of the $k$-th matching pair. According to the error $e_k$, the least squares problem is constructed for the set of matching pairs, which can be expressed as:

$$\min_{R,t} \sum_{k=1}^{m} \| (p_k - (Rp'_k + t)) \|^2, \tag{13}$$

where $m$ is the number of matching pairs in the set. Equation (13) can be simplified to the following two equations:

$$\min_{R,t} \sum_{k=1}^{m} \| q_k - Rq'_k \|^2, \tag{14}$$

$$t = p - Rp', \tag{15}$$

where $p$ and $p'$ are the centroids of the two point sets $P$ and $P'$, respectively, and $q_k$ and $q'_k$ are the de-centroid coordinates, which are expressed as:

$$q_k = p_k - p, q'_k = p'_k - p'. \tag{16}$$

Next, the optimal rotation matrix R is solved by SVD. The covariance matrix is constructed as follows:

$$Cov(P, P') = \sum_{k=1}^{m} q_k q'^{T}_k = \sum_{k=1}^{m} (p_k - p)(p'_k - p')^{T}. \tag{17}$$

Apply SVD on the covariance matrix:

$$Cov(P, P') = U\Sigma V^{T}, \tag{18}$$

where $\Sigma$ is a diagonal matrix composed of singular values. $U$ and $V$ are the matrices of the left and right singular vectors, respectively, and they are both diagonal matrices. When $Cov(P, P')$ is full rank, the rotation matrix $R$ can be solved by the following equation:

$$R = UV^{T}. \tag{19}$$

After obtaining the rotation matrix $R$, the translation vector $t$ can be obtained by (15).

## 4. Experiment

Experiments were conducted to verify the effectiveness and accuracy of the proposed method on plant point cloud registration. The computing platform is an Intel Core i5-10500 CPU which is made by the Intel Corporation of Santa Clara, CA, USA. Point clouds are generated by COLMAP [38] using sequences of images and the registration programs are developed based on the open-source library PCL.

Figures 7a and 8a are the actual scene photos of two plants (tree-1 and tree-2). According to the characteristics of the point cloud captured by UAV and a terrestrial capture platform, the point cloud of each tree is extracted into two parts. Figures 7b and 8b show the point cloud from the perspective of the terrestrial capture platform, which is called the target point cloud. Figures 7c and 8c are the point clouds captured by the UAV above trees, which are called source point clouds. Five sets of experiments with different overlap ratios were performed for each tree. The task of registration is to find the transformation between the source point cloud and the target point cloud. The overlap percentage between each set of point clouds to be registered is between 30% and 75%, which refers to the percentage of points in the overlap area to the total number of points. The similarity of the experimental data is reduced by randomly removing 20% of the points. In addition, voxel filtering also changes the coordinates of some points.

In order to verify the effectiveness, the proposed method is compared against SAC-IA and SAC-IA + ICP [20]. The registration results of the two plants are shown in Figures 7d–f and 8d–f, respectively. It can be seen from Figures 7d and 8d that when the overlap rate between point clouds is low, SAC-IA often gives results with large errors, and its registration success rate is low. As can be seen from #1 and #2 in Figure 7e, ICP fails when the coarse registration does not provide an approximately accurate initial pose. Similar results appear in #6 and #8 to #10 in Figure 8e. Compared with SAC-IA and ICP, our method performs better without obvious deviation, as shown in Figures 7f and 8f. In the case of a low overlap rate, our method can provide a more accurate transformation.

To quantitatively evaluate the effectiveness of point cloud registration, the root-mean-square error (RMSE) is used to measure the spatial distance error between the estimated transformed point cloud and the ground truth, which is defined as:

$$RMSE = \sqrt{\frac{\sum_{l=1}^{M} \parallel \boldsymbol{R}\boldsymbol{p}_l + \boldsymbol{t} - \boldsymbol{q}_l \parallel^2}{M}}. \tag{20}$$

where $\boldsymbol{R}$ and $\boldsymbol{t}$ represent the true values of the rotation matrix and translation vector, respectively, $\boldsymbol{p}_l$ and $\boldsymbol{q}_l$ represent points on the point cloud before and after the registration, and $M$ represents the number of points.

The RMSEs of the point cloud registration shown in Figures 7d–f and 8d–f are listed in Table 1. It can be seen that the errors of SAC-IA and ICP in experiments #1, #2, #6, and #8 to #10 are large, which is consistent with Figures 7 and 8. When the RMSE of a registration is greater than 10 cm, the registration is a failure, and it is meaningless to measure its accuracy. Ignoring errors larger than 10 cm, the mean values of the RMSEs of SAC-IA, SAC-IA + ICP, and the proposed method are, respectively, 0.38 cm, 0.012 cm, and 0.0027 cm. The mean RMSE of our method is only 0.71% and 22.5% of SAC-IA and SAC-IA + ICP, respectively. The proposed method, with higher rates of success and accuracy, outperforms the other two methods.

In practical applications, due to the influence of the surrounding environment, the constructed point cloud contains noise that directly affects the point cloud registration accuracy. To further verify the noise resistance of the proposed method, we added Gaussian noise with zero mean and standard deviation three times the resolution (RS) to all point clouds, and the results are shown in Figure 9. The definition of resolution RS is as follows:

$$RS = \frac{1}{n}\sum_{i=1}^{n} \parallel p_i - p_{in} \parallel, \tag{21}$$

where $n$ is the number of nearest neighbors of $p_i$, and $p_{in}$ is one of the nearest neighbors of $p_i$.

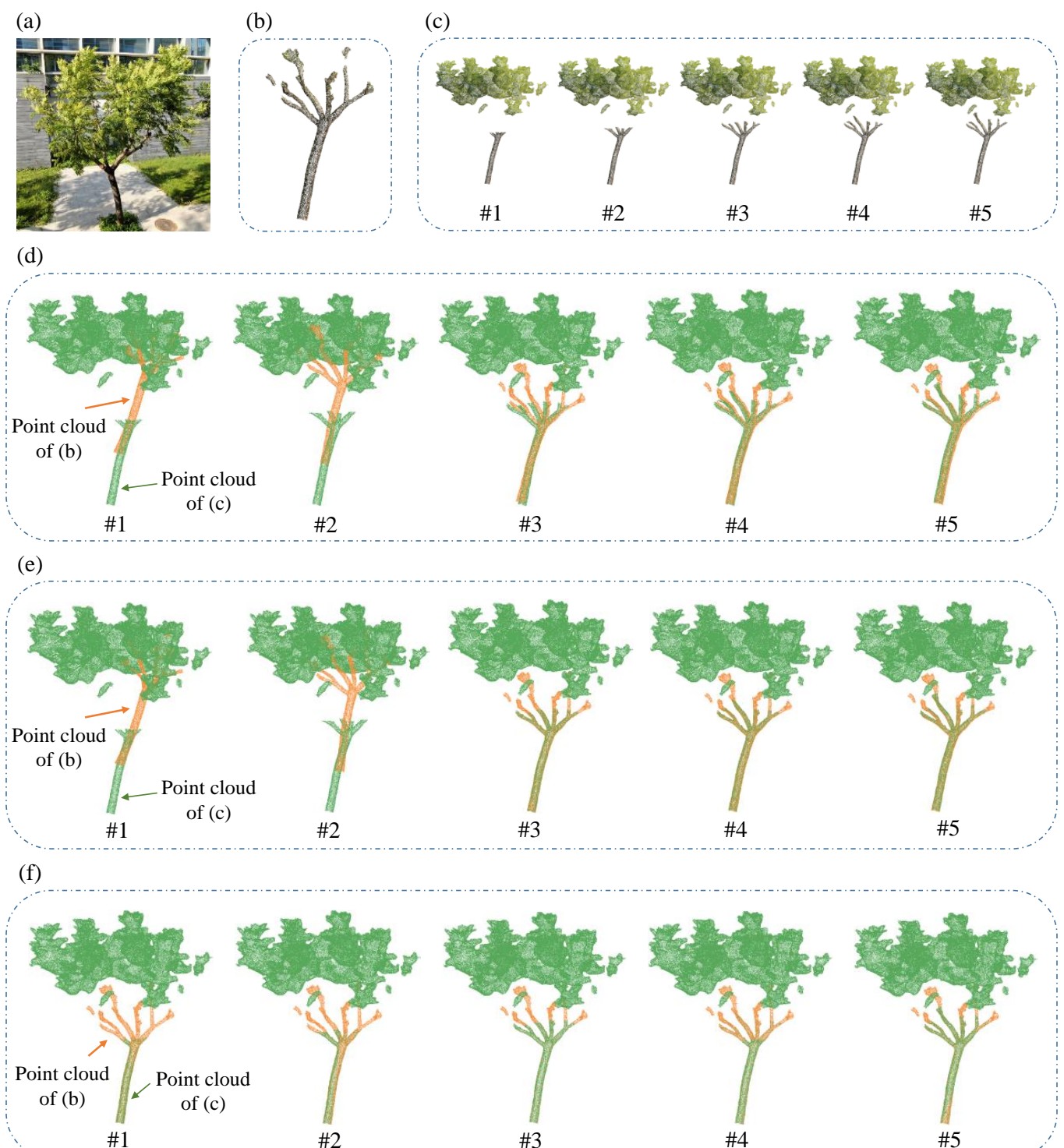

**Figure 7.** The inputs and the registration results of tree-1: (**a**) the actual scene photo of tree-1; (**b**) point cloud obtained by terrestrial capture platform; (**c**) point cloud obtained by UAV; (**d**–**f**) registration results of SAC-IA, SAC-IA + ICP, and the proposed method, respectively. Experiments are numbered from #1 to #5.

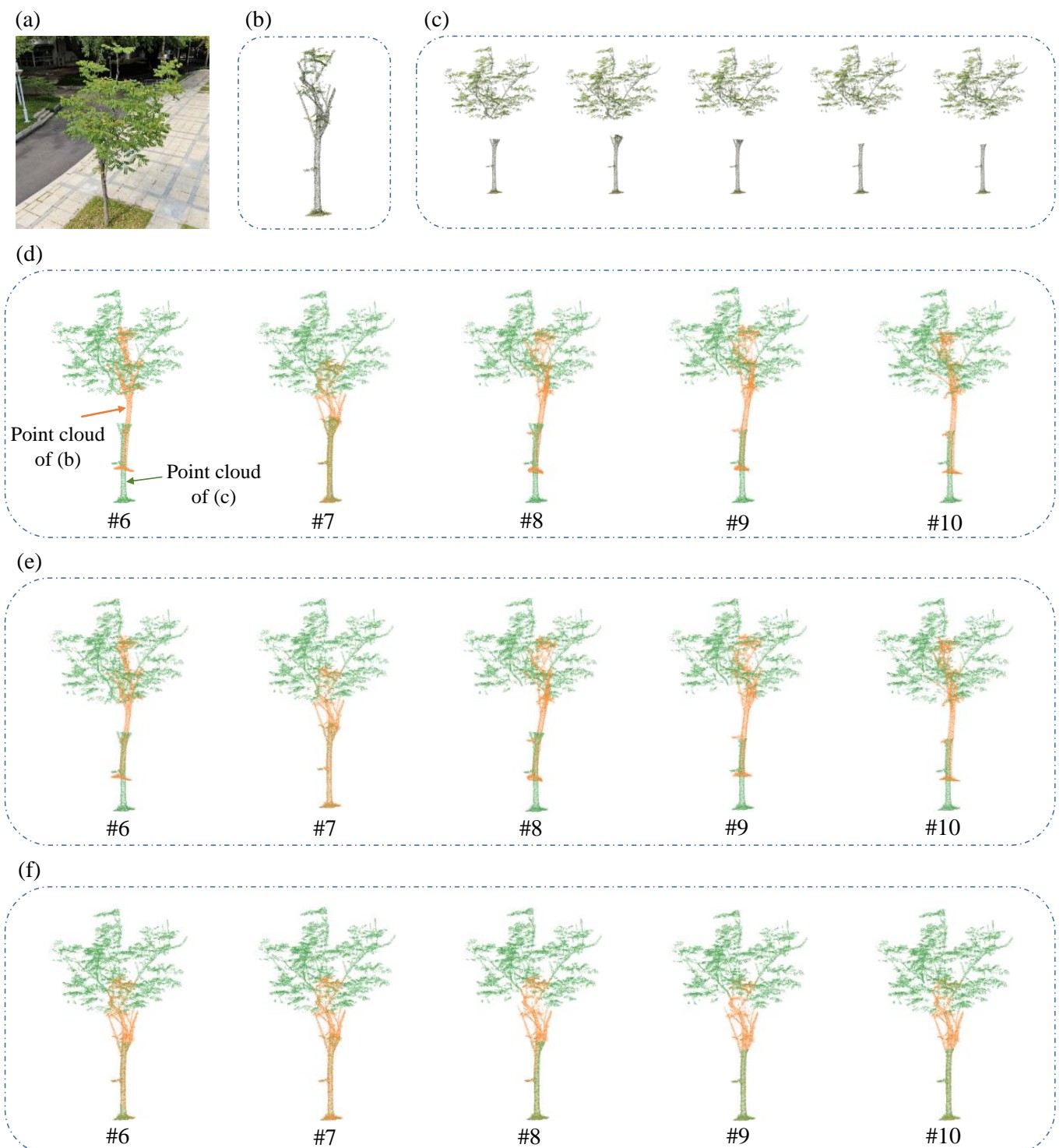

**Figure 8.** The inputs and the registration results of tree-2: (**a**) the actual scene photo of tree-1; (**b**) point cloud obtained by terrestrial capture platform; (**c**) point cloud obtained by UAV; (**d**–**f**) registration results of SAC-IA, SAC-IA + ICP, and the proposed method, respectively. Experiments are numbered from #6 to #10.

**Table 1.** The RMSEs of the point cloud registration shown in Figures 7d–f and 8d–f.

| Point Cloud Umber | SAC-IA (cm) | SAC-IA + ICP (cm) | Proposed (cm) |
|---|---|---|---|
| #1 | 210.9800 | 197.4900 | 0.0037 |
| #2 | 92.6800 | 93.3900 | 0.0064 |
| #3 | 0.8700 | 0.0036 | 0.0000 |
| #4 | 0.2500 | 0.0080 | 0.0002 |
| #5 | 0.3100 | 0.0370 | 0.0069 |
| #6 | 65.2200 | 64.3700 | 0.0022 |
| #7 | 0.0870 | 0.0007 | 0.0007 |
| #8 | 53.5400 | 57.6000 | 0.0052 |
| #9 | 56.0100 | 61.3400 | 0.0009 |
| #10 | 48.1900 | 50.8000 | 0.0007 |
| Mean[#] | 0.3793 | 0.0123 | 0.0027 |

Mean[#]: The mean of the RMSE values that are less than 10 cm.

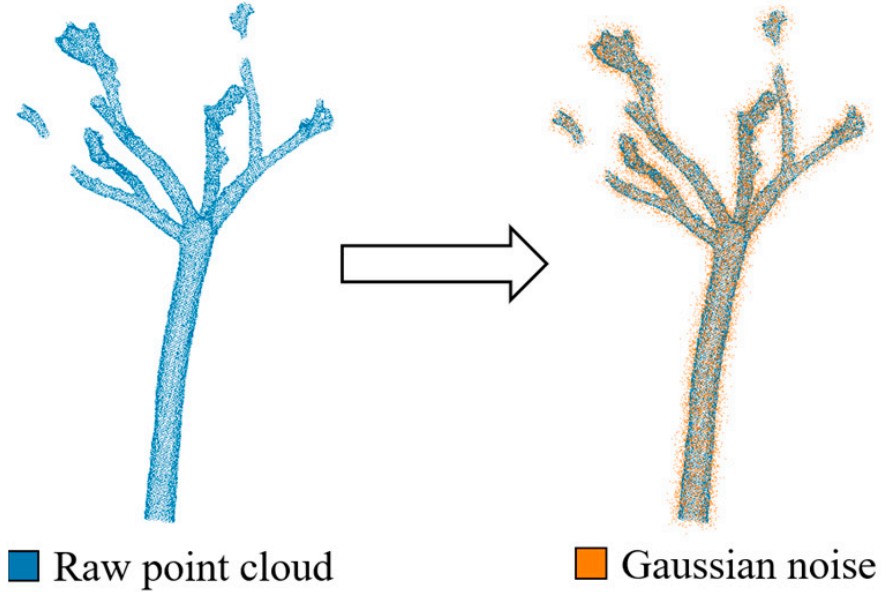

**Figure 9.** The raw point cloud and the point cloud with Gaussian noise.

Ten comparison experiments were carried out with the point clouds with Gaussian noise, which were processed with SAC-IA, SAC-IA + ICP, and the proposed method. The noisy point clouds were registered, and the results are shown in Figures 10 and 11. With noise interference, registration is more difficult. SAC-IA and ICP become more prone to registration failure, as shown in Figure 10a,b. However, the proposed method still maintains good registration performance and is able to align the source and target point clouds, as shown in Figures 10c and 11c.

The RMSE is also used as the precision measurement of the anti-noise registration effect, and the results are shown in Table 2. The RMSEs of the proposed method are smaller than those of SAC-IA + ICP while maintaining a high success rate. Ignoring the experiments with RMSE greater than 10 cm, the mean values of the RMSEs of SAC-IA, SAC-IA + ICP, and the proposed method are 1.12 cm, 0.33 cm, and 0.0039 cm, respectively. The mean RMSE of our method is only 0.35% and 1.18% of SAC-IA and SAC-IA + ICP, respectively. The results show that when registering noisy point clouds, the proposed method aligns more tightly than other methods. The proposed method not only has a higher success rate and accuracy but also is more robust.

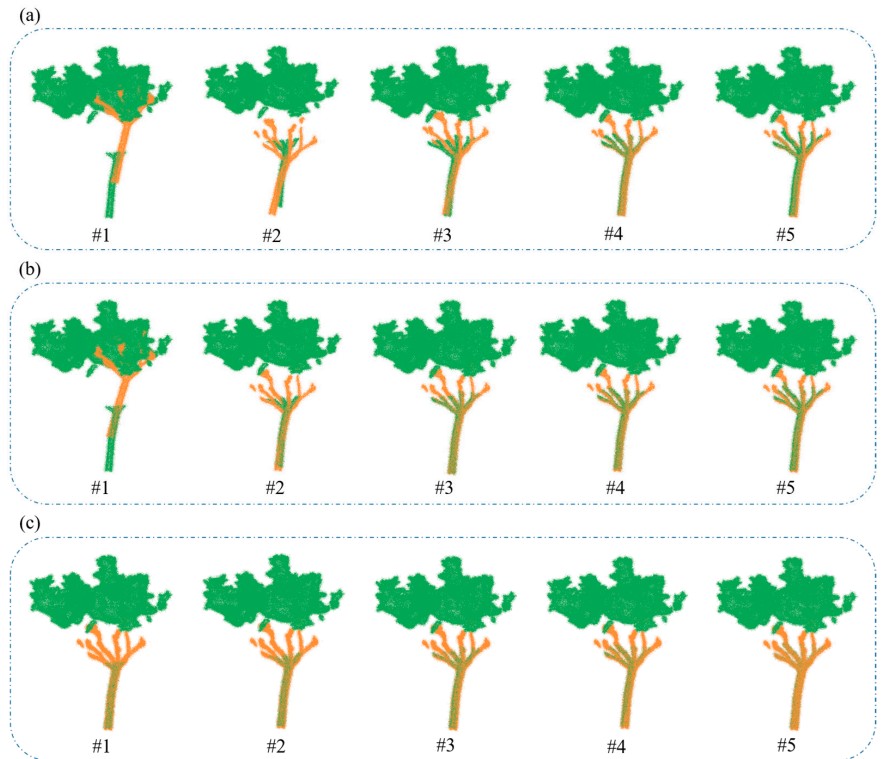

**Figure 10.** The registration results of tree-1 with added noise: (**a**–**c**) the registration results obtained from SAC-IA, SAC-IA + ICP, and the proposed method, respectively. Experiments are numbered from #1 to #5, corresponding to those shown in Figure 7.

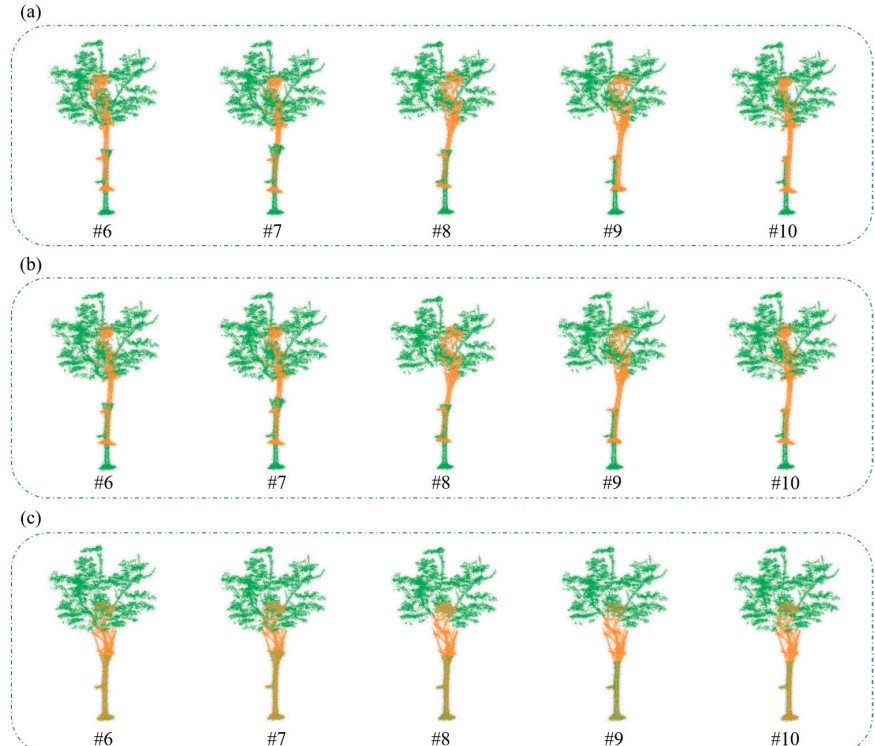

**Figure 11.** The registration results of tree-2 with added noise: (**a**–**c**) the registration results obtained from SAC-IA, SAC-IA + ICP, and the proposed method, respectively. Experiments are numbered from #6 to #10, corresponding to those shown in Figure 8.

**Table 2.** The RMSEs of the point cloud registration shown in Figures 10 and 11.

| Point Cloud Umber | SAC-IA (cm) | SAC-IA + ICP (cm) | Proposed (cm) |
|:---:|---:|---:|---:|
| #1 | 126.5200 | 118.6100 | 0.0084 |
| #2 | 3.6700 | 0.8800 | 0.0037 |
| #3 | 0.1500 | 0.0720 | 0.0033 |
| #4 | 0.2500 | 0.1500 | 0.0067 |
| #5 | 0.4000 | 0.2200 | 0.0014 |
| #6 | 43.8200 | 47.7500 | 0.0009 |
| #7 | 41.1500 | 50.2100 | 0.0020 |
| #8 | 60.1600 | 61.0900 | 0.0022 |
| #9 | 47.4700 | 56.1400 | 0.0008 |
| #10 | 45.5200 | 54.8500 | 0.0097 |
| Mean$^{\#}$ | 1.1175 | 0.3305 | 0.0039 |

Mean$^{\#}$: The mean of the RMSE values that are less than 10 cm.

## 5. Conclusions

In order to obtain a more complete point cloud set of tall plants, a low overlap rate plant point cloud registration method has been developed in this work. After acquiring the partial point cloud from the vision sensor, the geometric features in the plant point cloud are described by FPFH. This local reference frame-based feature allows registration not to be limited by the initial pose. The Bhattacharyya distance, which has good properties, is used as a similarity measure to score the correlation between features. By selecting a reliable matching pair set, the least squares problem is constructed to solve the rigid transformation between point clouds.

The method proposed is suitable for general scenarios without initial pose guessing, and is also suitable for point clouds with low overlap rates. Our method is resource efficient, and its registration is completed in one run instead of iterations. It also does not require training the model like deep learning methods. After experimental verification, the method can better complete the registration of plant point clouds with a low overlap rate. It also outperforms SAC-IA and SAC-IA + ICP, with a higher success rate and accuracy.

The proposed method is helpful to improve the efficiency and accuracy of plant phenotype monitoring. The extracted phenotypes, including height, diameter at breast height and canopy size, are important indicators to measure plant growth. Thus, it has great value for practical applications in agriculture and forestry fields.

Future work will be focused on automatically adjusting the parameters of statistical filtering used in different scenarios. Multi-sensor data fusion also will be considered to combine visual features and LiDAR information to further improve the stability and reliability of large-scale point cloud registration for accurate 3D reconstruction of plants.

**Author Contributions:** Conceptualization, Y.P. and S.L.; funding acquisition, Y.P.; methodology, S.L.; validation, H.W. and G.C.; writing—original draft, Y.P., S.L. and H.W.; writing—review and editing, Y.P., H.W. and G.C. All authors have read and agreed to the published version of the manuscript.

**Funding:** This work was supported by the National Natural Science Foundation of China (Grant No. 51905351) and the Science and Technology Planning Project of Shenzhen Municipality, China (Grant No. JCYJ20190808113413430).

**Data Availability Statement:** The readers who would like to acquire our research data (multi-view images of trees) can contact the email: yeping.peng@szu.edu.cn.

**Conflicts of Interest:** The authors declare no conflict of interest.

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
