# Peer review of "Point Cloud Registration Based on Fast Point Feature Histogram Descriptors for 3D Reconstruction of Trees"

_remotesensing, doi:10.3390/rs15153775_

Round 1

Reviewer 1 Report

Good approach to the work and a good list of quotations, but for the problem posed, you should use control points should be used for ground-based point acquisition and the same points for aerial point acquisition, i.e. georeferencing in the point cloud.

Perhaps it would be useful to look at the possibilities with other programmes, both free or paid. For instance, Cloudcompare and Metashape.

Author Response

We thank the reviewer for the valuable comments. The responses to the reviewer comments are given below.

Comment 1: Good approach to the work and a good list of quotations, but for the problem posed, you should use control points should be used for ground-based point acquisition and the same points for aerial point acquisition, i.e. georeferencing in the point cloud.

Response: We would like to express our gratitude for your comment. It is indeed true that ground-based and aerial points are frequently employed as georeferencing points for point cloud registration. However, we propose a novel FPFH-descriptor-based point cloud registration approach that eliminates the need for control points while still achieving accurate registration results. This approach leverages the geometric features of different local point clouds to acquire their spatial relationships. As a result, it effectively circumvents the errors in point cloud matching that may arise due to imprecise or inadequate control points.

Comment 2: Perhaps it would be useful to look at the possibilities with other programmes, both free or paid. For instance, Cloudcompare and Metashape.

Response: We had carefully assessed the capabilities of CloudCompare and Metashape, as stated in the manuscript. However, our suggestion was to employ these programs specifically for pre-processing tasks, such as background removal. For the registration stage, our proposed method is developed based on the literature's reported strategy of sample consensus initial alignment (SAC-IA) and iterative closest point (ICP) algorithms. Therefore, in order to compare the performance, we conducted experiments that involved SAC-IA, SAC-IA-ICP, and our proposed method.

Reviewer 2 Report

Dear authors, 

I would like to share some questions that I got after reading your manuscript.

- I suppose the g value (proportionality coefficient) have values in the range (0-100 or 0.0-1.0), but I would like to know which value do you finally choose for it, or maybe you change it in every run?. You said it is related with number of neighbor points and the exclusion criteria. Please, explain further this, what number? what exclusion criteria?

- You processed the data cloud (background removal, denoising and downsampling); also cropped the target tree manually, and finally, introduce outliers generated by yourself into the point cloud, to avoid some problems (mismatching and registration failures, ease of computation, etc.). How much do you consider that could affect the behavior of your proposal method with real data (i mean, not adjusted)?

Not any other comment by this moment. 

Best regards, 

Author Response

We thank the reviewer for the valuable comments. The responses to the reviewer comments are given below.

Comment 1: I suppose the g value (proportionality coefficient) have values in the range (0-100 or 0.0-1.0), but I would like to know which value do you finally choose for it, or maybe you change it in every run? You said it is related with number of neighbor points and the exclusion criteria. Please, explain further this, what number? what exclusion criteria?

Response: Thank you for the comment. Statistical filtering is a technique based on the Gaussian distribution, where the majority of data points are expected to fall within a specific range around the mean value. Any data points that significantly deviate from the mean can be considered outliers. To detect these outliers, a commonly used approach involves evaluating the average distance of each point to its k-nearest neighbors using a global threshold. In Equation (1), the parameter 'g' serves as a proportionality coefficient that adjusts the threshold. If a point's average distance to its k-nearest neighbors exceeds the threshold, it is identified as an outlier and removed from consideration. Determining the appropriate 'g' value depends on the distribution of noise points and the complexity of the point cloud. Empirically, the recommended range for 'g' is between 1.0 and 3.0. A higher 'g' value results in fewer points being removed, but more noise being retained. This additional information has been included in Section 3.1 of the revised manuscript.

Comment 2: You processed the data cloud (background removal, denoising and downsampling); also cropped the target tree manually, and finally, introduce outliers generated by yourself into the point cloud, to avoid some problems (mismatching and registration failures, ease of computation, etc.). How much do you consider that could affect the behavior of your proposal method with real data (i mean, not adjusted)?

Response: The preprocessing steps for the point cloud, which involve background removal, denoising, and downsampling, do not alter the object points themselves. Background removal aims to separate the target tree from the surrounding environment, leading to reduced computational requirements for object matching. Denoising is employed to eliminate noise generated by imaging sensors, thereby preventing the inclusion of irrelevant outlier points in the constructed point clouds. Additionally, downsampling is performed to reduce the number of points, enhancing the efficiency of the registration process. As a result, these preprocessing steps are implemented to enhance both the accuracy and efficiency of point cloud registration.

To assess the robustness of the proposed method, artificial outliers are intentionally added to the constructed point clouds. Although manually generated points may not fully replicate the characteristics of real spatial points derived from image feature points, but it aids in quantitatively analyzing the stability of the point cloud registration method across different applications. Furthermore, experiments were conducted using real data without artificial outliers, as depicted in Figure 7 and Figure 8. The results clearly demonstrate that our method achieves satisfactory registration accuracy with practical point clouds.

Reviewer 3 Report

This manuscript proposed a method for the 3D reconstruction of trees, which uses point cloud registration based on fast point feature histogram descriptors. The authors obtained the overall structural features of trees including the trunk and crown by a combination of low-altitude remote sensing (an unmanned aerial vehicle) and a terrestrial capture platform (a mobile robot), and fast point feature histogram (FPFH) was used to describe the geometric features in the plant point cloud. The results show that the method proposed can registrant the plant point clouds with a low overlap rate, and has a higher accuracy. However, the manuscript still contains many points and needs to be revised.

1.       The first time an abbreviation appears in the manuscript, please provide the full name or a note, such as “LiDAR” in line 34, etc.

2.       In the introduction, it is strongly recommended that the authors add the significance of 3D reconstruction, the current common methods, and applications, reflecting the great value of 3D reconstruction. The current manuscript only includes the 3D reconstruction of plants. However, "3D reconstruction is widely used in medical, mechanical, agricultural, urban planning and other fields", some supplementary references are as follows:

https://doi.org/10.1177/00202940211033881

https://doi.org/10.1016/j.compag.2022.107210

https://doi.org/10.1016/j.aei.2021.101501

https://doi.org/10.1016/j.autcon.2021.104092

3.       When the authors cite references, most of them are simply displayed in the paper. I suggest that the authors relate these references to the work of this paper, for example, how certain studies have influenced their work.

4.       In the section introductions, please use Arabic numerals instead of Roman numerals, consistent with the numbering method in the manuscript.

5.       For the two methods SAC-IA and SAC-IA + ICP used for comparison, what is the difference in the main idea compared with the proposed method? What caused such a significant increase in accuracy? Why did the authors choose these methods for comparison? How about other methods?

6.       Please clearly highlight how your work advances the field from the present state of knowledge and you should provide a clear justification for your work. The impact or advancement of the work can also appear in the conclusion.

7.       It is suggested that the author add some prospects for future research directions.

It is suggested that the authors have someone competent in the English language and the subject matter of your manuscript go over the manuscript and polish it.

Author Response

We thank the reviewer for the valuable comments. The responses to the reviewer comments are given below.

Comment 1: The first time an abbreviation appears in the manuscript, please provide the full name or a note, such as “LiDAR” in line 34, etc.

Response: Thank you for your comment. We have thoroughly reviewed the manuscript and have made sure to include the full names or explanatory notes for all abbreviations in the revised version.

Comment 2: In the introduction, it is strongly recommended that the authors add the significance of 3D reconstruction, the current common methods, and applications, reflecting the great value of 3D reconstruction. The current manuscript only includes the 3D reconstruction of plants. However, "3D reconstruction is widely used in medical, mechanical, agricultural, urban planning and other fields", some supplementary references are as follows:

https://doi.org/10.1177/00202940211033881

https://doi.org/10.1016/j.compag.2022.107210

https://doi.org/10.1016/j.aei.2021.101501

https://doi.org/10.1016/j.autcon.2021.104092

Response: The importance of 3D reconstruction has been highlighted in the Introduction section, and relevant references have been cited to support this significance.

Comment 3: When the authors cite references, most of them are simply displayed in the paper. I suggest that the authors relate these references to the work of this paper, for example, how certain studies have influenced their work.

Response: Additional explanations have been incorporated in the Introduction section, specifically addressing the relationship between the cited references and our proposed method. These revisions provide further clarity on how the reported methods align with our approach.

Comment 4: In the section introductions, please use Arabic numerals instead of Roman numerals, consistent with the numbering method in the manuscript.

Response: We have thoroughly reviewed the manuscript and made sure that all numerals are now formatted in Palatino Linotype, in accordance with the formatting style specified by Remote Sensing.

Comment 5: For the two methods SAC-IA and SAC-IA + ICP used for comparison, what is the difference in the main idea compared with the proposed method? What caused such a significant increase in accuracy? Why did the authors choose these methods for comparison? How about other methods?

Response: Point cloud registration, which relies on geometric feature matching, is an effective approach for achieving accurate registration results without the need for control or reference points. Among the widely used algorithms for point cloud registration are the sample consensus initial alignment (SAC-IA) and iterative closest point (ICP) algorithms. In order to enhance registration accuracy, a hybrid method (SAC-IA + ICP) was developed by combining SAC-IA and ICP. However, this approach requires proper initialization and may be susceptible to local optima. To address these challenges, we propose a correspondence estimation method that matches point clouds and calculates spatial transformations to facilitate registration. This technique leads to improved point cloud matching and registration accuracy. Detailed information regarding this approach can be found in Section 3.3 and Section 3.4 of the manuscript.

To evaluate the performance of our proposed method, we compare it with the SAC-IA and SAC-IA + ICP methods, which are selected as benchmarks for comparison.

In evaluating the performance of our proposed method, we compare it with the SAC-IA and SAC-IA + ICP methods, which are selected as benchmarks for comparison. While there are other point cloud registration methods available, such as the 3Dmatch method and learning-based approaches like PointNet and PointNet++, these methods were not included in the comparison due to their complex computations, reliance on large training datasets, and the need for high-end computer configurations. In contrast, our proposed method can be executed on conventionally configured computers, where learning-based methods often encounter difficulties. For reference, our computing platform consists of an Intel Core i5-10500 CPU with 8GB RAM (as indicated in Section 4).

Comment 6: Please clearly highlight how your work advances the field from the present state of knowledge and you should provide a clear justification for your work. The impact or advancement of the work can also appear in the conclusion.

Response: The highlights of this work are as follows:

  • We propose an association description and assessment method based on FPFH and Bhattacharyya distance. This method exhibits rotational invariance to point clouds and demonstrates improved accuracy for point clouds with noise and low overlap.
  • We develop a matching pair pruning strategy using RANSAC. By evaluating the correlation between matching pairs, incorrect matches are effectively removed, leading to an increased proportion of correct matching pairs.
  • Through a series of registration experiments on real-world tree point clouds and synthesized point clouds with Gaussian noise, we demonstrate the effectiveness of the proposed method.

Furthermore, in the Conclusion section, we highlight the advancements of this work:

The proposed method holds promise in enhancing the efficiency and accuracy of plant phenotype monitoring. The extracted phenotypes, such as height, diameter at breast height, and canopy size, serve as crucial indicators for measuring plant growth. As a result, this method holds significant practical value in the fields of agriculture and forestry.

Comment 7: It is suggested that the author add some prospects for future research directions.

Response: Our future work will primarily concentrate on automatically adjusting the parameters of statistical filtering to adapt to various scenarios. Additionally, we plan to explore the potential of multi-sensor data fusion, combining visual features and LiDAR information. This integration aims to enhance the stability and reliability of large-scale point cloud registration, ultimately facilitating accurate 3D reconstruction of plants. These research directions have been included in the Conclusion section of the revised manuscript.

Round 2

Reviewer 3 Report

The manuscript has been improved a lot according to the reviewers' comments. The authors carefully checked the whole manuscript and addressed all the comments seriously. I think the manuscript can be accepted in its present form.